# Characterization and In Vitro Biocompatibility of Two New Bioglasses for Application in Dental Medicine—A Preliminary Study

**DOI:** 10.3390/ma15249060

**Published:** 2022-12-18

**Authors:** Andra Clichici, Gabriela Adriana Filip, Marcela Achim, Ioana Baldea, Cecilia Cristea, Gheorghe Melinte, Ovidiu Pana, Lucian Barbu Tudoran, Diana Dudea, Razvan Stefan

**Affiliations:** 1Department of Propaedeutics and Dental Materials, Faculty of Dental Medicine, Iuliu Hatieganu University of Medicine and Pharmacy, 400012 Cluj-Napoca, Romania; 2Department of Physiology, Iuliu Hatieganu University of Medicine and Pharmacy, 400006 Cluj-Napoca, Romania; 3Departments of Pharmaceutical Technology and Biopharmaceutics, Iuliu Hatieganu University of Medicine and Pharmacy, 400606 Cluj-Napoca, Romania; 4Analytical Chemistry Department, Faculty of Pharmacy, Iuliu Haţieganu University of Medicine and Pharmacy, 400349 Cluj-Napoca, Romania; 5National Institute for Research and Development of Isotopic and Molecular Technologies, 400293 Cluj-Napoca, Romania; 6Faculty of Veterinary Medicine, University of Agricultural Sciences and Veterinary Medicine, 400372 Cluj-Napoca, Romania

**Keywords:** bioglasses, biocompatibility, human dysplastic keratinocytes, oxidative stress, DNA lesions

## Abstract

Bioactive glasses (BGs), also known as bioglasses, are very attractive and versatile materials that are increasingly being used in dentistry. For this study, two new bioglasses—one with boron (BG1) and another with boron and vanadium (BG2)—were synthesized, characterized, and tested on human dysplastic keratinocytes. The in vitro biological properties were evaluated through pH and zeta potential measurement, weight loss, Ca^2+^ ions released after immersion in phosphate-buffered saline (PBS), and scanning electron microscopy (SEM) coupled with energy dispersive spectroscopy (EDS) analysis. Furthermore, biocompatibility was evaluated through quantification of lactate dehydrogenase activity, oxidative stress, transcription factors, and DNA lesions. The results indicate that both BGs presented the same behavior in simulated fluids, characterized by high degradation, fast release of calcium and boron in the environment (especially from BG1), and increased pH and zeta potential. Both BGs reacted with the fluid, particularly BG2, with irregular deposits covering the glass surface. In vitro studies demonstrated that normal doses of the BGs were not cytotoxic to DOK, while high doses reduced cell viability. Both BGs induced oxidative stress and cell membrane damage and enhanced NFkB activation, especially BG1. The BGs down-regulated the expression of NFkB and diminished the DNA damage, suggesting the protective effects of the BGs on cell death and efficacy of DNA repair mechanisms.

## 1. Introduction

Bioactive glasses are modern materials with extensive use in regenerative medicine, including in dental medicine. In dentistry, bioactive glasses have been considered for bone grafts—to substitute bone and to promote osteogenesis—and for bone regeneration, as well as for soft tissue substitutes, drug delivery systems, antimicrobial agents for endodontic and periodontic treatment, and topical disinfectants during endodontic procedures [1,2]. The bioglasses (BGs) used for coating dental implants can enhance osseointegration into alveolar bone. Additionally, BGs can be used in dentifrices or topical applications to treat gingivitis, to improve dentin hypersensitivity, and as abrasive materials in dental air abrasion machines [3]. In 1969, Larry Hench designed the first bioglass composition called 45S5 Bioglass, which is considered a biocompatible product that is capable of bonding to living tissues and stimulating osteogenesis through the release of active ions [3]. However, 45S5 Bioglass is not an ideal bioglass. Although 45S5 Bioglass can form a bone-like hydroxyapatite layer in contact with living tissues, it has drawbacks, such as the cytotoxicity induced by the high sodium content and elevated pH environment. It also has a decreased sintering ability, making the creation of porous 3D scaffolds difficult. In fact, this first bioglass presents a high dissolution rate, poor sintering ability, and a cytotoxic effect [4].

For our study, we started from the well-known concept that modifying the BG composition can affect the biocompatibility and degradation rate, as well as the ease of material processing [5]. By adding ions, the BG structure can be modified, improving the properties of such materials for their use in biomedical applications, especially in dentistry. Metal ions, such as magnesium, strontium, manganese, iron, zinc, and silver, have been shown to increase the mechanical and biological properties of bioactive glasses, facilitating healing and tissue regeneration processes [6]. Resin composites with fluoride-containing bioactive glass (BAG) have been shown to enhance dentin re-mineralization and eliminate enzymatic degradation at the dentin interface [7].

The novelty of this study comes from the fact that we synthetized two new BGs—one containing boron (BG1) and another containing boron and vanadium (BG2)—which were characterized and tested regarding their biocompatibility. We chose boron due to its important role in structural integrity preservation through maintaining normal cell function, stimulating the release of cytokines and growth factors and promoting regeneration of the extracellular matrix. It also has been shown to possess anti-inflammatory, antibacterial, and antiviral effects in various in vitro and in vivo models [8], and has presented positive effects regarding wound-healing, healthy bone development, regeneration, and angiogenesis [9]. As for vanadium, we chose it because it is present in the bones and vanadate groups possess phosphate-like functions in the regulation of physiological processes [10]. In experimental studies, the addition of vanadium to a collagen matrix increased biocompatibility as well as the adhesiveness of the product, and induced the differentiation of osteoblasts [11].

Considering the above, we started from the premise that the addition of boron and vanadium into the BG composition could increase the biocompatibility and improve their effects in biological systems. Studying the possible toxicity of these products is a mandatory step in the evaluation of their effects, taking into account the fact that the response of a tissue depends on the type of material and the dose used. Their use in dentin re-mineralization and restorations, due to their remarkable antibacterial properties and/or ability to stimulate angiogenesis and bone regeneration, represent innovative qualities of these materials which are important in the context of dentistry. Therefore, we aimed to characterize the two new synthesized BGs and evaluate their biocompatibility in order to assess their suitability for further biomedical applications in dentistry. The null hypothesis was that there is no difference between the two BGs, based on their composition, with respect to bioactivity and in vitro toxicity.

For this purpose, the BGs were characterized by *X*-ray photoelectron spectroscopy (XPS) and *X*-ray diffraction (XRD); furthermore, after immersion in simulated fluids and phosphate-buffered saline (PBS), their pH, zeta potential, and weight loss were measured, as well as the concentration of Ca^2+^ ions released, followed by scanning electron microscopy (SEM) coupled with EDS analysis. The in vitro biocompatibility of the BGs was evaluated according to lactate dehydrogenase (LDH) activity, an oxidative stress assessment, transcription factor expression, and DNA lesion quantification.

## 2. Materials and Methods

### 2.1. Reagents

The substances used for synthesizing the bioactive glasses (H_3_BO_3_, CaF_2_, and V_2_O_5_), along with the Bradford reagent, 2-thiobarbituric acid, glutamine, DMEM, 10% fetal bovine serum (FBS), hydrocortisone, antibiotics, and antifungals were purchased from Sigma Aldrich, GmbH (Darmstadt, Germany). PBS buffer was acquired from Lonza (Brussels, Belgium) and standard Ca^2+^ solutions from Fluka (Frankfurt, Germany). Phosphorylated histone H_2_AX (pS139; γH_2_AX) was purchased from Stressgen Bioreagents Corporation (Victoria, BC, Canada). Antibodies against nuclear factor kappa-light-chain-enhancer of activated B cells (NF-kB) and pNF-KB were obtained from Cell Signaling Technology, Inc. (Danvers, MA, USA) and glyceraldehyde 3-dehydrogenase (GAPDH) was acquired from Santa Cruz Biotechnology (Santa Cruz, CA, USA).

### 2.2. Preparation and Characterizations of the BGs

For this study, two bioactive glasses—BG1: 100 [2B_2_O_3_·CaF_2_] and BG2: 0.5V_2_O_5_·99.5 [2B_2_O_3_·CaF_2_]—were developed using fundamental materials of high purity (H_3_BO_3_, CaF_2_, and V_2_O_5_). The materials were weighed in certain proportions, mixed in an agate pestle for homogenization for 15 min, and then placed in a melting furnace at a melting point of 1150 °C in air. The melts were brought rapidly below the crystallization temperature and pressed between stainless steel plates, obtaining cylindrical sample pieces. The samples were then ground into agate crucibles and passed through a 2.0–4.0 mm sieve, in order to obtain controlled sized powders. The composition analysis of the solid BGs composition was performed by XPS analysis and *X*-Ray diffractometry. In order to investigate the in vitro bioactivity, the two BGs were immersed in simulated fluids (PBS) for different periods of time (1/4, 1, 3, 7, and 14 days). The pH and zeta potential were assessed up to 10 days, while the weight loss measurements and the concentration of Ca^2+^ ions released after immersion in PBS, followed by scanning electron microscopy (SEM) coupled with EDS analysis, were performed up to 14 days. The in vitro biocompatibility of the tested materials was evaluated according to LDH activity, oxidative stress, transcription factor expression, and DNA lesion quantification. 

### 2.3. In Vitro Evaluation of Bioactivity

For this study, fragments of BG1: 100 [2B_2_O_3_·CaF_2_] and BG2: 0.5V_2_O_5_·99.5 [2B_2_O_3_·CaF_2_] with a diameter of 2–4 mm were used. During the in vitro evaluation of bioactivity, the BG samples immersed in PBS were maintained at 37 °C in a climate chamber (Binder, Germany). A calibrated pH meter (Mettler Toledo, Columbus, OH, USA) and a Zetasizer Nano ZS90 (Malvern, UK) were used for measurement of the pH and the zeta potential, respectively.

## 3. pH and Zeta Potential Measurement

The BG samples were immersed in PBS 0.0067M phosphate (pH 7.27, zeta potential 0.963 mV) at a ratio of 0.2 g per 100 mL PBS (Ohaus Analytical balance). Both the pH and zeta potential were measured at different time intervals up to 10 days (1, 2, 3, 6 h; 1, 2, 4, 7 and 10 days), directly from samples containing BG fragments. [12].

### 3.1. Weight Loss and In Vitro Release of Calcium Ions

For weight loss and in vitro release of calcium ions, 0.2 g of BG samples were immersed into 100 mL PBS and maintained at 37 °C. At certain time intervals (6 h; 1, 3, 7, and 14 days), the dispersions were centrifuged at 5000 rpm and the Ca^2+^ ion concentration was determined in the supernatant. The BG samples were washed with distilled water, dried for 4 h in an oven at 90 °C, and then weighed. The cumulative weight loss (ΔM/Mo) was calculated as a function of immersion time (ΔM = Mo − M, where Mo represents the initial mass and M is the mass at time t). All tests were performed in triplicate [12].

### 3.2. Calcium Ion Detection by a Potentiometric Method

Ca^2+^ ions were determined by a potentiometric method using a combined electrode HI4104 from Hanna Instruments (Rome, Italy), consisting of a Ca^2+^ ion-selective electrode (ISE), an Ag/AgCl reference electrode, and a Consort C830 potentiometer (Brussels, Belgium). The tests were performed in volumes of 10 mL under continuous stirring using an Arex magnetic stirrer (Velp Scientifica, Usmate Velate, Italy) at room temperature. The first step was to draw a calibration plot using Ca^2+^ standard solution in the range 0.25–20 µg mL^−1^ (6.238−499 µM) in 0.1 M KCl starting from a standard solution of 10 g L^−1^ (Fluka, Frankfurt, Germany), obtaining the calibration plot considering the potential (E) of the solution as a function of −log_10_ molar concentration of Ca^2+^ : y (mV) = −26,275 × (−log_10_ [Ca^2+^]) + 56,052, with a correlation coefficient of R^2^ = 0.992. The sample solutions were diluted 1:1 with a solution of 0.2 M KCl, obtaining sample solutions containing 0.1 M KCl in the end in order to keep the ionic forces of the standard solutions used for the calibration plot constant. All samples were potentiometrically analyzed, performing three measurements for each sample, and the average concentrations (µg/mL) of Ca^2+^ for each moment were calculated. The variation in time of Ca^2+^ concentration was followed after 6 h (1/4 day) and on days 1, 3, 7, and 14.

### 3.3. SEM Analysis of BGs

For characterization by SEM, the BG samples were immobilized on aluminum rods using double-sided adhesive sheets (Electron Microscopy Sciences, Hatfield, MA, USA). The samples were then spray-coated with a 10-nm gold layer in a Polaron E-5100 sprayer (Polaron Equipment Ltd., Watford, UK) in the presence of argon (45 s at 2 kV and 20 mA). Ultrastructural images were obtained at 10 kV with different magnifications using a Hitachi SU8230 electron microscope (Hitachi, Japan).

## 4. In Vitro Evaluation of BGs Toxicity

### 4.1. Cell Cultures and Preparation of Sample BGs Extract

Human dysplastic oral keratinocytes (DOK) (ECCAC 94122104) were purchased from Sigma Aldrich (Heidelberg, Germany), and were used in passage 31–32. The culture medium was DMEM, supplemented with 2 mM glutamine, 10% FBS (fetal bovine serum), 5 pg/mL hydrocortisone, antibiotics, and antifungals; all reagents were purchased from Sigma Aldrich, Co (Heidelberg, Germany). Samples of the experimental biomaterials were incubated in the culture medium at a concentration of 2 g/mL and at a temperature of 37 °C for 24 h, following the procedures detailed in ISO 10993-12/2012 [13]. Then, the extract obtained was sterile filtered and used immediately for in vitro experiments. The sample, prepared as indicated by the producer, was incubated with medium in the same conditions as the experimental biomaterials.

### 4.2. Cell Viability Assay

To assess cell viability, the CellTiter 96^®^ AQueous Non-Radioactive Cell Proliferation Assay (Promega Corporation, Madison, WI, USA) was used. Dysplastic keratinocytes were cultured for 24 h at a density of 10^4^/well in 96-well plates (TPP, Trasadingen, Switzerland). Then, the cells were exposed for 24 h to the extracts in different concentrations (1, 0.5, 0.25, and 0.125). The experiments were performed in triplicate, and untreated cell cultures were used as controls. Formazan, a compound synthesized by viable cells, was measured with a colorimetric method at 540 nm, using an ELISA plate reader (Tecan, Männedorf, Switzerland). The results are expressed as % of untreated control, where a decrease in cell viability below 70% was considered to indicate cytotoxic effects.

### 4.3. Preparation of Cell Lysates

For the preparation of cell lysates, DOK cells were inoculated on Petri dishes at a density of 10^4^/cm^2^ for 24 h, then exposed for 24 h to undiluted sample extracts. Untreated cells were used as controls. After exposure, the cells were washed and lysed according to a previously described method [14]. The Bradford method [15] was used to determine the protein concentration in the lysate, according to the manufacturer’s specifications (Biorad, Hercules, CA, USA), using bovine albumin serum as a standard.

### 4.4. Lactate Dehydrogenase (LDH) and Oxidative Stress Assessment

In order to assess the toxicity of the tested compounds, lactate dehydrogenase (LDH) activity and oxidative stress were evaluated. LDH activity, which can quantify cell membrane damage, was evaluated in culture media using a spectrophotometric method, according to a previously reported technique [13]. LDH is expressed in units of enzymatic activity (nMol NAD/mL/min). Malondialdehyde (MDA), a marker of lipid peroxidation, was evaluated using a fluorometric method with 2-thiobarbituric acid [16]. The results are expressed as nmole/mg protein.

### 4.5. Evaluation of Transcription Factors and DNA Lesions

For the evaluation of mechanisms involved in the toxicity of the BGs, the transcription factor NFkB and its activated form pNFkB, as well as γH2AX, a marker of DNA double-strand breaks, were assessed by Western blotting. Briefly, cell lysates (20 µg protein/lane) were separated by electrophoresis on SDS PAGE gels and transferred to polyvinylidenedifluoride membranes using a Biorad Miniprotean system (BioRad, Hercules, CA, USA). Blots were blocked and then incubated with antibodies against NFkB, pNFkB, and γH2AX, then further washed and incubated with corresponding secondary peroxidase-linked antibodies. The amount of protein was measured by the Bradford method [15]. The proteins were detected using Supersignal West Femto Chemiluminiscent substrate (Thermo Fisher Scientific, Rockford, IL, USA), and a Gel Doc Imaging system equipped with an XRS camera and the Quantity One analysis software (Biorad, Hercules, CA, USA). Glyceraldehyde 3-phosphate dehydrogenase (GAPDH; Trevigen Biotechnology Gaithersburg, MD, USA) was used as a protein loading control.

## 5. Statistical Analysis

The statistical significance regarding the differences between treated and control (untreated) cells was evaluated by a two-way ANOVA and paired Student’s *t*-test, followed by the Bonferroni post-hoc test using the GraphPad Prism version 5.00 software for Windows (GraphPad Software, San Diego, CA, USA, www.graphpad.com, accessed on 20 October 2022). A *p*-value <0.05 was considered to indicate statistical significance. All reported data are expressed as the mean of triplicate measurements ± standard deviation (SD).

## 6. Results

### 6.1. XPS Measurements and X-ray Diffraction

In order to obtain information about the chemical compositions before immersion, *X*-ray photoelectron spectroscopy (XPS) measurements were performed on powdered samples of the two BGs. As an example, the XPS spectra of Ca 2*p*, F 1*s*, B 1*s*, and V 2*s* core-level lines corresponding to the sample with vanadium (BG2) are illustrated in Figure 1. The observed chemical states, according to the NIST X-ray Photoelectron Spectroscopy Database [NIST Standard Reference Database Number 20, National Institute of Standards and Technology, Gaithersburg, MD, USA, 20,899 (2000), doi:10.18434/T4T88K] were that B 1*s* indicated B_2_O_3_, while F 1*s* indicated metal fluorides. In the case of V 2*s*, the peak observed at 632.67 eV may originate from V_2_O_5_. The Ca 2*p* line has limited use in determining chemistry, while the 2*p* (3/2) line positioned at 349.5 eV can be attributed to CaF_2_, but the shift to higher BE energy indicates the presence of Ca–OH bonds, which may be present in the sample near the surface [17]. As XPS is a surface analysis technique, the calculated sample stoichiometry cannot be fully expected due to surface defects. As in our previous studies [18], Ar sputtering was performed for 1 h at 3000 V, equivalent to a depth of approximately 5.3 nm below the surface. The line positions, FWHM, normalized areas, and At. % composition at ~5.3 nm depth are listed in Table 1, for both BG1 and BG2. 

For a better correlation between sample composition and their behavior in biological fluids, the ratio of atom pairs was calculated, as listed in Appendix A. Through *X*-ray diffraction (XRD), no crystalline phases were detected; the diffraction patterns being large and noisy is characteristic of vitreous and homogeneous samples (Appendix A).

### 6.2. Bioactivity Assay of BGs

In view of future applications in biology, the pH value and zeta potential of two samples after immersion of the BGs in PBS were monitored for 10 days. The weight loss and the concentration of Ca^2+^ ions released were measured at ¼, 1, 3, 7, and 14 days after immersion and, at the same time intervals, SEM coupled with EDS analysis was performed. 

Generally, BGs immersed in biological fluids undergo a biodegradation process, as a result of which ion exchange with the environment takes place and a hydroxyapatite (HA) layer [Ca_11_(PO_4_)_6_(OH)_2_] forms on solid surfaces [12]. This ion exchange process induces changes in the pH and zeta potential in the environment, characteristics that reflect the biodegradation of these materials and the extent of the transformation. The evolution of the pH in the case of the studied BGs is shown in Figure 2.

According to Figure 2, immersion of BG1 and BG2 into PBS induced a slight increase in pH values over the ten days of study, from 7.27 to 7.65. In the first two days, the increase was more marked, reaching values around 7.52; then, the values grew slowly until day 10. Under the indicated experimental conditions, between the two studied BGs, no significantly different pH values were recorded, indicating that the presence of vanadium oxide did not lead to noticeable changes in the concentration of hydrogen/hydroxyl ions. Lower pH values were observed compared with other studies. In the case of borate BG micro-fibers, the pH values reached up to 9.25, and for a silicate BG, they reached up to 8.65 [12]. This difference could be due to the lack of sodium ions in the studied BG, as the sodium released into the environment can cause a significant increase in pH. The zeta potential after immersion in PBS at 37 °C was also monitored for 10 days, and the results are shown in Figure 3A,B.

The zeta potential increased very rapidly in the first hour after immersion, then dropped slightly, with a minimum at 6 h (Figure 3B). A further increase was noticed, which was faster until day 2, and then slowed down (Figure 3A). This evolution demonstrated the existence of ion exchange between the BGs and the immersion medium. Based on the composition of the two BGs, the main ions transferred could be Ca^2+^ and (BO_3_)^3−^, according to other similar studies [19]. The sudden increase in zeta potential at the beginning of the immersion could be explained by the rapid degradation of boric BGs. The presence of boron in the network of bioactive glasses determined a different structure from that of silicate-based glasses, as boron has only three valences compared with Si (which has four); this aspect leads to faster biodegradation [20]. The decrease in zeta-potential values occurring between 2–6 h after immersion (Figure 3B) may be related to the formation of hydroxyapatite. The presence of vanadium oxide did not lead to important differences in the evolution of zeta potential of the two studied BGs.

For both studied BGs, the accumulative weight loss increased with the immersion time in PBS, without significant differences between the two samples (Figure 4). In the first 6 h after immersion, there was a significant weight loss per day (5.32%/day), representing 10.5% of the total loss. In the next 18 h, the changes were very small. Following that, the interval from 1–3 days registered the most accelerated rate of degradation (17.6%/day; 34.12% of the total weight loss). In the interval from 3–14 days, the intensity of the process decreased, but evolved with almost constant speed, with a weight loss of 5.4%/day. At the end of the study, after 14 days, the total weight loss was between 12.50–13%. According to these data, the presence of vanadium oxide did not change the rate of weight loss and biodegradation of the BGs. During the analyzed time interval, the concentration of Ca^2+^ ions registered an increasing evolution (Figure 5). Thus, in the first 6 h after immersion there was a significant release of Ca^2+^ ions, representing between 45–55% of the total increase recorded. For BG1, the calcium release was faster, reaching a maximum value after 3 days, which remained constant even after 14 days. In the case of BG2, the release of Ca^2+^ ions was slower, but an increasing trend was maintained over the 14 days. Based on these results, we can state that vanadium oxide decreased the rate of Ca^2+^ ion release. Compared with other studies performed on borate BGs [12], the Ca^2+^ ion concentrations were lower after 6 h and after the first day, likely due to the fact that, in our study, the BG particles had a larger diameter and a smaller surface area, which could explain their slower biodegradation. 

### 6.3. SEM Analysis of BGs

For better visualization of the interaction of the BG samples with PBS, images obtained by SEM at the same magnification for all immersion time intervals (1/4, 1, 3, 7, and 14 days) are shown in Figure 6. The SEM images indicate the presence of irregular formations randomly arranged on the initially smooth surface of the glass (BG1 in 1d), while the second sample analyzed under the same conditions (BG2 1d) showed a wavy surface full of inhomogeneities. Analysis of the two samples at 3 days after immersion revealed significant surface changes on both samples, which were similar in appearance, without being able to accurately separate the shape and consistency of the layer deposited in either case. As can be seen from Figure 6, the formed layer became more and more compact as the immersion time increased. In Figure 6, it can be seen that BG2 reacted with the fluid faster than BG1. In order to identify the composition of the structures formed on the two sample surfaces, elemental X-ray analysis (EDS) was performed. From the elemental analysis (Table 2) obtained from capturing the energy dispersed by surfaces at different points, it was observed that elements were deposited on the surface of the two samples from the immersion liquid. However, in the case of BG1, the quantity remained constant over time for all the identified elements. In the case of BG2, the analyzed elements covered the surface of the material better, with an observable increase of their quantity. At 14 days the existing gaps in the surface were covered, suggesting the formation of a surface deposit. Traces of aluminum (Al) and silicon (Si) were also found, indicating slight corrosion of the crucibles during melting. The carbon (C) signal came from the surface on which the two samples were fixed, which also dispersed the incident *X*-ray beam. No traces of vanadium (V) were detected, as it was introduced into the starting substances in very small quantities in order to reduce the toxicity of the samples, allowing for the possibility of their use in biomedical applications.

The elemental analysis obtained by the combined SEM-EDS technique also revealed elements from the biological fluid (e.g., Na and P) on the surface of the samples without being able to state that their mass increased over time, as the surfaces under analysis were different at the time of analysis.

## 7. In Vitro Toxicity

### 7.1. Cell Viability Tests

The viabilities of the DOK cells exposed to the sample extracts in increasing dilutions are shown in Figure 7. The two experimental biomaterials were well-tolerated by the cells, without decreasing their viability below the toxicity limit. BG2 presented the best viability values. A decrease in cell viability was observed only after incubation with high doses of the BGs. For LDH activity and oxidative stress assessments and for western blot analysis a concentration of 2 µg/mL BGs was used. 

### 7.2. LDH Activity and Oxidative Stress Assessment

In Figure 8, the LDH activities in the supernatant of DOK cells exposed to the two studied BGs are shown (Figure 8A). LDH activity increased significantly in the supernatant of DOK cells exposed to the two bioactive glasses, compared with the control untreated cells (*p* < 0.001). No significant difference was observed in terms of LDH activities or cell membrane integrity between the two BGs (*p* > 0.05).

As shown in Figure 8B, the MDA levels in cell lysates obtained after exposure to bioactive glasses increased statistically significantly in cells exposed to the two BGs, compared with untreated cells (*p* < 0.001). No statistically significant difference was found between the two bioactive glasses tested (*p* > 0.05).

### 7.3. Evaluation of Transcription Factors and DNA Lesions

The Western blot analysis (Figure 9) demonstrated that expression of the constitutive form of NFkB decreased in the presence of both BGs. The tested compounds induced activation of NFkB, especially BG1 (*p* < 0.001), suggesting the presence of activators of NFkB in the bioglass composition. The studied bioactive glasses did not induce DNA damage, with the most protective of them being BG1 due to the lack of vanadium oxide in its composition (*p* < 0.001).

## 8. Discussion

Bioactive glasses have a multitude of applications in medicine and dentistry; for example, in the repair and regeneration processes of different types of tissue (especially bone tissue), as well as re-mineralization of tooth surfaces in order to treat dental hypersensitivity [3,21]. These properties of BGs derive from their ability to stimulate the formation of bone due to the release of ions capable of inducing osteogenic stem cell differentiation [21]. BGs have the ability to release ions and to modify their properties, depending on their composition. Therefore, in recent years, the study of BGs has developed into an important direction of research, taking as an objective the modification of their ionic composition in order to improve their physical properties, mechanical properties, and chemical stability, and thus improve their impact on living tissues [22].

In this study, the properties and biocompatibility of two bioactive glasses—both with boron trioxide (B_2_O_3_) as a glass network former and one with vanadium pentoxide (V_2_O_5_) as an impurity—were evaluated. The behavior of BG1 and BG2 in simulated fluids (PBS) was assessed in terms of the ability to release calcium ions, measurement of weight loss, pH and zeta potential, and evaluation of their appearance under SEM. XPS analysis before immersion was also performed in order to identify the ions in the solid BG composition. Their biocompatibility was estimated in vitro using DOK cells, according to cell viability, ability to generate free radicals, and evaluation of the expression of transcription factors and DNA damage. The results obtained indicated the good interaction of both BGs with the simulated fluid, with faster release of calcium and boron in the medium and formation of large irregular deposits on the glass surface, particularly after BG2 immersion in PBS. The two BGs had no effect on cell viability and DNA, and down-regulated NFkB expression in DOK but induced redox imbalance and cell membrane damage in parallel with NFkB activation, especially BG1. These effects suggested that the BGs are not completely inert, as they may induce lipid peroxidation of membrane lipids through NFkB activation and proinflammatory state. However, their good behavior in simulated environments, lack of cytotoxicity, and protective effect on DNA are important arguments for their use in medical practice. The null hypothesis was accepted, as there was no difference between the bioactivity and in vitro effects of both BGs. However, further studies are required to decipher the mechanisms that lead to oxidative stress and, consequently, to membrane damage, as well as the importance of these aspects regarding living organisms.

Through this study, we attempted to find answers to some questions. The first was: what are the benefits of adding boron and/or vanadium into the composition of the newly synthesized BGs? It is well-known that boron is a microelement that plays a significant role in preserving structural integrity, thus maintaining normal cellular function due to its ability to stimulate the release of cytokines and growth factors, as well as promoting regeneration of the extracellular matrix. It also possesses anti-inflammatory, antibacterial, and antiviral abilities [8]. These actions have been supported by numerous experimental evidence [23]. Vanadium is normally present in bones, and vanadate groups have phosphate-like functions in regulating physiological processes [10]. In in vitro experiments, the addition of vanadium to a collagen matrix increased the biocompatibility and adhesiveness of the product, and induced the differentiation of osteoblasts [11].

In view of the above, we started from the premise that the presence of boron or vanadium in the BG composition could increase the biocompatibility and improve the effects of these compounds in biological systems. Therefore, the study of their possible toxicity was considered a mandatory step in evaluating their biological effects, taking into account that the response of a tissue depends both on material type and the dose used. Second, the characterization and evaluation of the physico-chemical properties of the newly synthesized BGs was conducted, as a required step in the evaluation of a new material. Therefore, the behavior of the two BGs in simulated fluids, including measurement of pH and zeta potential, weight loss, and in vitro release of Ca^2+^ ions, supplemented by SEM examination, X-ray powder diffraction (XRD), and elemental analysis by X diffraction, was evaluated. XPS was performed on the solid BGs in order to better identify their composition. The degradation of BGs immersed in simulated fluids, ion exchange with the immersion fluid, the change in the pH of the medium over time, and the formation of a hydroxyapatite layer on solid surfaces were quantified. The results obtained did not show significantly different behavior between the two BGs tested. The pH changes recorded after immersion in PBS showed no statistically significant differences, indicating that the presence of vanadium oxide did not lead to noticeable changes in the concentration of hydrogen or hydroxyl ions. In our experiment, the pH recorded was lower than in studies carried out with boric BG, in the form of cotton wool (the pH was up to 9.25) and a silica BG (the pH was up to 8.65) [12]. This difference could be due to the lack of sodium ions in the tested BGs, as sodium released into the environment can cause a significant increase in pH. In addition, although ion exchange between the BGs and the immersion medium were highlighted, the presence of vanadium oxide did not lead to important differences between the studied BGs. However, the decrease in value of the zeta potential within 2–6 h after immersion suggested hydroxyapatite formation on the solid surfaces. For both BGs, the cumulative mass loss increased with the time of immersion, without any significant difference between the two samples. The highest weight loss was noticed within the interval of 1–3 days; moreover, at the end of the study, the total weight loss was between 12.50–13%. The presence of vanadium oxide did not change the rate of mass loss and biodegradation of the BG2. During the analysis period, the concentration of Ca^2+^ ions released increased, which was faster in BG1, with a maximum at 3 days, and slower in BG2 with vanadium oxide, which presented an increasing trend throughout the experiment. It seems that vanadium oxide slowed down the rate of release of Ca^2+^ ions, consistent with other studies in the literature [12]. SEM images indicated the appearance of irregular formations arranged randomly on the smooth surface of the BG with boron trioxide, while the addition of vanadium oxide increased the reaction time with the fluid, leading to the appearance of a wavy and inhomogeneous surface with more compact layers as the interaction increased. Through elemental X-ray analysis, traces of boron, calcium, and fluoride were detected but not vanadium, likely due to the very low amounts of vanadium initially introduced into the composition of BG2. Third, the toxicity assessment of the newly synthesized BGs was conducted, which is extremely important for biomedical applications in dentistry. The toxicity was tested on DOK cell lines through assessment of cell viability, LDH activity (as a marker of cell membrane damage), MDA level, the expression of transcription factors NFkB and pNFkB, and γH2AX formation (as a marker of DNA damage). In terms of cell viability, the newly synthesized BGs did not influence cell viability at low doses but increased cell toxicity at high doses. The viability of cell lines was maintained at over 70% viable cells, suggesting that both BGs are not toxic in vitro. Data from the literature suggests that boron may have toxic effects in a concentration-dependent manner. Thus, boron-containing BGs have demonstrated toxic effects on certain cell types, including bone cells [24]. In human osteoblasts, boron concentrations above 1000 ng/mL led to short-term toxic effects, but these effects were not observed under long-term exposure [25]. Boron at a concentration of 6.25 mg/mL has been shown to have toxic effects on dental pulp stem cells [26]. Studies [27] have also shown that nanoparticles loaded with vanadium trioxide exerted toxic effects on lung endothelial and epithelial cells due to the conversion of vanadium trioxide to vanadium pentoxide. Some researchers [28] have explained the toxicity of vanadium in terms of the interaction with other metals from the medium, such as selenium or iron. To assess the biocompatibility of the BGs, the cell membrane damage, free radical production, transcription factors, and DNA damage were quantified. LDH activity—a parameter that quantifies cell membrane damage—was increased after treatment with the two BGs, in parallel with the increase of MDA levels, suggesting the production of free radicals as a possible mechanism of lipid peroxidation and membrane damage. The constitutive expression of NFkB decreased in cells exposed to BGs, while BG1 induced NFkB activation, suggesting the possible role of oxidative stress as an inducing factor for NFkB activation. Both BGs were not toxic to DNA and the γH2AX formation decreased, demonstrating the efficiency of DNA damage repair mechanisms. The evaluation of oxidative stress, as an indicator of biocompatibility, started from the observations of some studies showing that some dental materials—even new-generation ones—may produce oxidative stress and cytotoxicity in some cell lines [29]. There are even studies that have claimed that some materials widely used in dentistry, including restorative materials, have more important pro-oxidant and cytotoxic effects than amalgam [30]. Even materials considered to be biocompatible have been shown [31] to induce oxidative stress and inflammation, sounding the alarm regarding the need for robust assessments of the toxicity of dental materials. Oxidative stress is defined as an imbalance between the production of reactive oxygen or nitrogen species and the body’s antioxidant, enzymatic, or non-enzymatic defenses. Reactive species can attack proteins, nucleic acids, carbohydrates, and lipids, and can change their structure to lose their specific functions [32]. In our study, the oxidative stress produced by the two BGs was highlighted by a marker of lipid peroxidation, demonstrating the attack of free radicals on lipids. Free radicals themselves have a very short life and are sometimes difficult to detect. For this reason, products obtained from the interaction of free radicals with macromolecules are more frequently measured; most commonly, malondialdehyde is used, which is a result of lipid peroxidation [33,34]. For BGs with antioxidant effect, it has been shown that the occurrence of gingival disorders secondary to dental caries was attenuated [35]. For BGs with vanadium oxides, some studies have demonstrated pro-oxidant effects, depending on the type of vanadium oxide used and the cellular compartment in which it accumulates [36,37]. In order to evaluate the toxic effects on DNA, the γH2AX foci were assessed by Western blot analysis. In studies performed on periodontal fibroblasts C165 using the comet test [38], the genotoxic effects of bioactive nanoglasses and Novobone microglasses were revealed, depending on the concentration and the time of exposure. In our study, both BGs did not induce DNA lesions, suggesting their protective effects against cell death and the efficacy of DNA repair mechanisms.

The limitations of the current study derive from the fact that we were unable to use primary cultures of keratinocytes obtained from the oral mucosa, as the proliferation of this type of keratinocyte is difficult to realize, even for experienced professionals, and they tend to change phenotype after a few passages. This did not allow all desired experiments to be performed so we chose to use DOK cells.

In future, we would like to compare these two newly synthesized BGs with those that are already used in dental medicine. We also intend to integrate these BGs into restorative dental materials in order to determine whether, once incorporated, the properties of the BGs are preserved and if they still have the ability to release ions. Additionally, we aim to evaluate whether the biocompatibility of restorative dental materials can be improved by the addition of the BGs, through both in vitro and in vivo studies.

## 9. Conclusions

In this study, we demonstrated that our two proposed BGs present the same behavior in simulated fluids, characterized by high degradation, fast release of calcium and boron in the environment (especially from BG1), and increased pH and zeta potential. Both BGs reacted with the fluid (particularly BG2), with irregular deposits covering the glass surface. The sudden rise in pH after day 8 could be explained by the rapid degradation of boric BG, and the likely deposition of hydroxyapatite on their surface. The concentration of Ca^2+^ in the medium showed an increasing evolution, especially within the first 6 h after immersion, due to the release of these ions; notably, the vanadium oxide in the BG composition slowed the release of calcium ions into the environment. SEM images of the studied BGs indicated the formation of irregular structures arranged randomly on their surface, with the deposited layer becoming more and more compact as the immersion time was prolonged. In vitro studies showed that normal doses of the BGs were not cytotoxic to DOK cells, while high doses reduced cellular viability. Both BGs induced oxidative stress and cell membrane damage, and enhanced NFkB activation. Additionally, the BGs down-regulated the expression of NFkB and diminished DNA damage, suggesting protective effects of the BGs on cell death and the efficacy of DNA repair mechanisms. The results obtained are promising, but future studies are required to decipher the mechanisms involved in oxidative lesions of the cell membrane and the importance of these lesions for the health of people and animals. Their use in dental medicine, especially their inclusion in different dental materials, can provide a beneficial therapeutic option, considering their protective effect on important mechanisms of cell death; however, further studies are still required.

## Figures and Tables

**Figure 1 materials-15-09060-f001:**
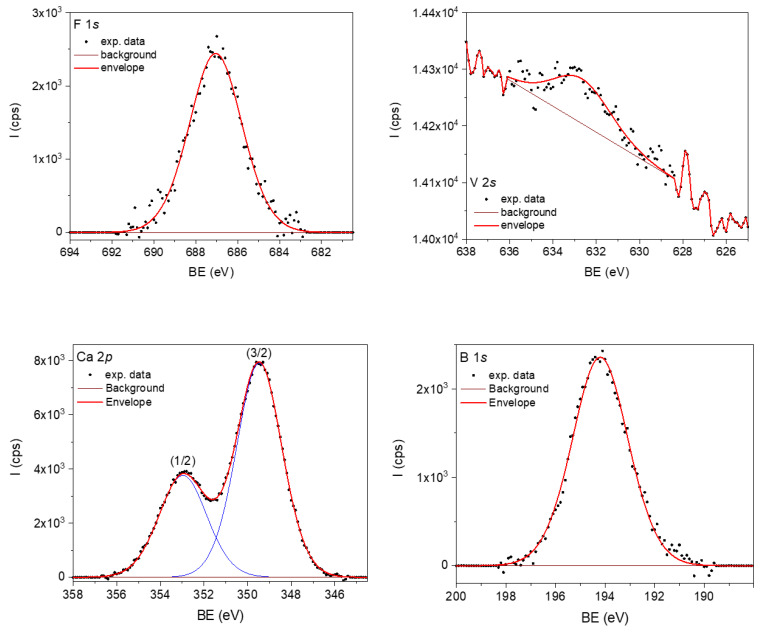
XPS spectra of F 1*s*, V 2*s*, Ca 2*p*, and B 1*s* core-level lines corresponding to sample BG2.

**Figure 2 materials-15-09060-f002:**
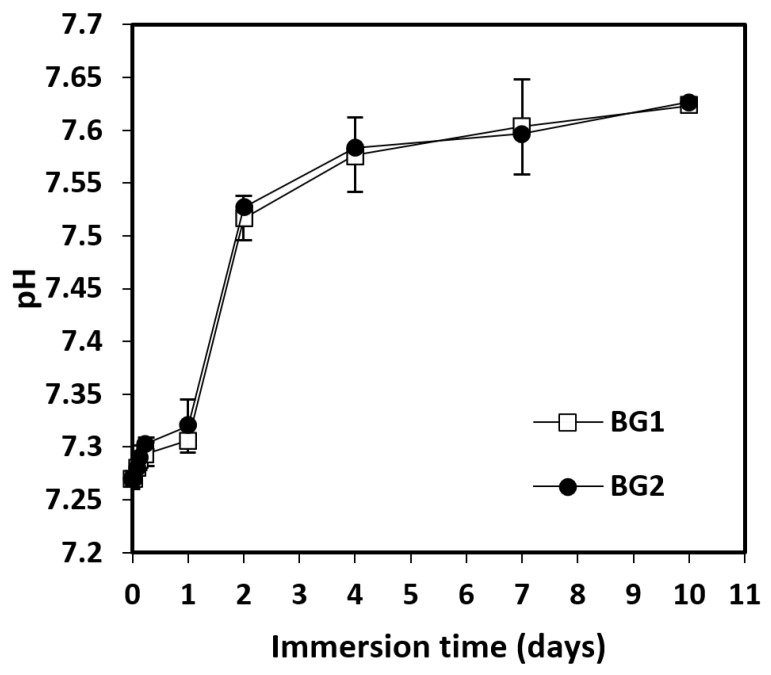
pH values of BGs immersed in PBS at 37 °C as a function of immersion time (BG1, 100[2B_2_O_3_·CaF_2_]; BG2, 0.5V_2_O_5_·99.5[2B_2_O_3_·CaF_2_]). Mean ± SD, *n* = 3.

**Figure 3 materials-15-09060-f003:**
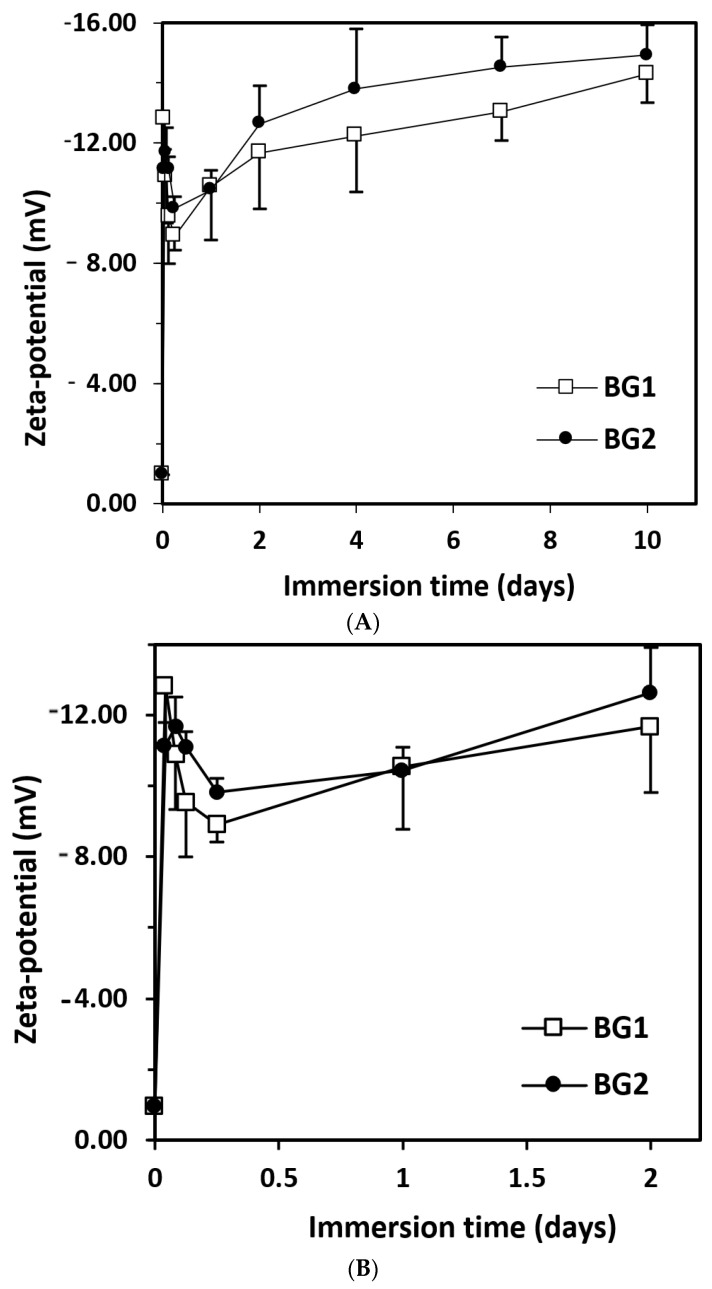
Zeta potential of BGs immersed in PBS at 37 °C as a function of immersion time (BG1, 100[2B_2_O_3_·CaF_2_]; BG2, 0.5V_2_O_5_·99.5 [2B_2_O_3_·CaF_2_]). (**A**) Zeta potential for 10 days; (**B**) Zeta potential for 2 days. Mean ± SD, *n* = 3.

**Figure 4 materials-15-09060-f004:**
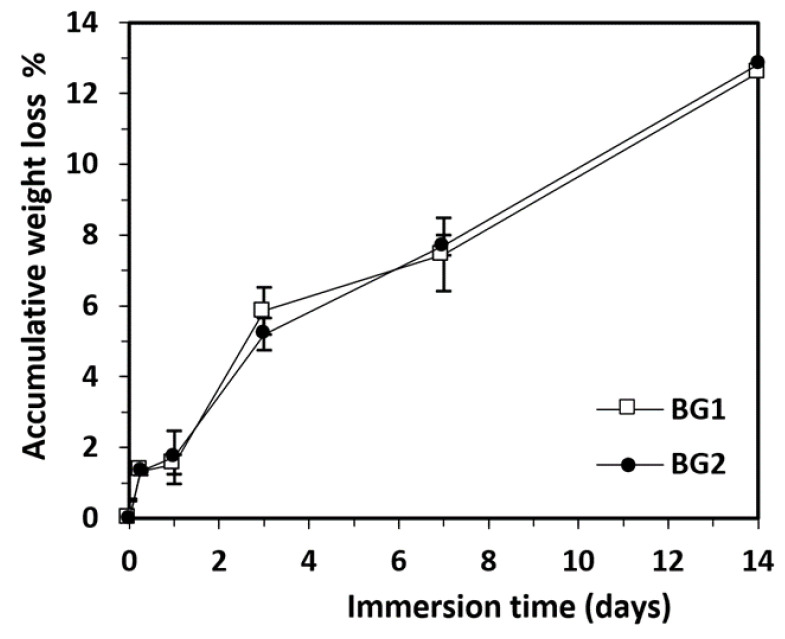
Accumulative weight loss of BG samples immersed in PBS at 37 °C as a function of immersion time (BG1, 100[2B_2_O_3_·CaF_2_]; BG2, 0.5V_2_O_5_·99.5[2B_2_O_3_·CaF_2_]). Mean ± SD, *n* = 3.

**Figure 5 materials-15-09060-f005:**
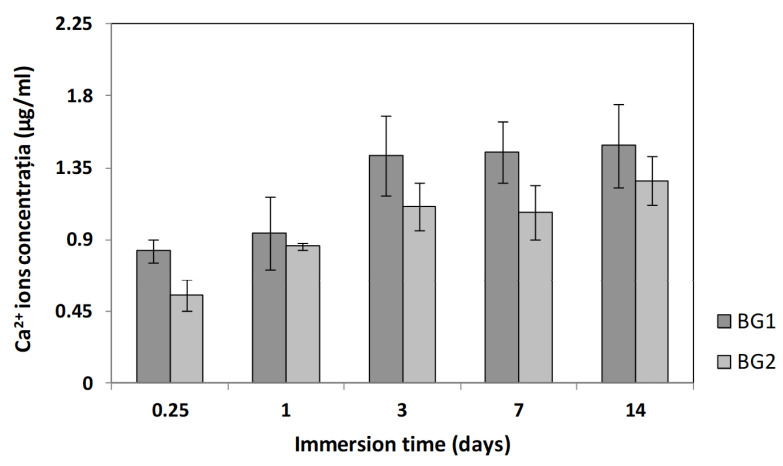
Ca^2+^ ions released from BGs immersed in PBS at 37 °C as a function of immersion time (BG1, 100[2B_2_O_3_·CaF_2_]; BG2, 0.5V_2_O_5_·99.5[2B_2_O_3_·CaF_2_]). Mean ± SD, *n* = 3.

**Figure 6 materials-15-09060-f006:**
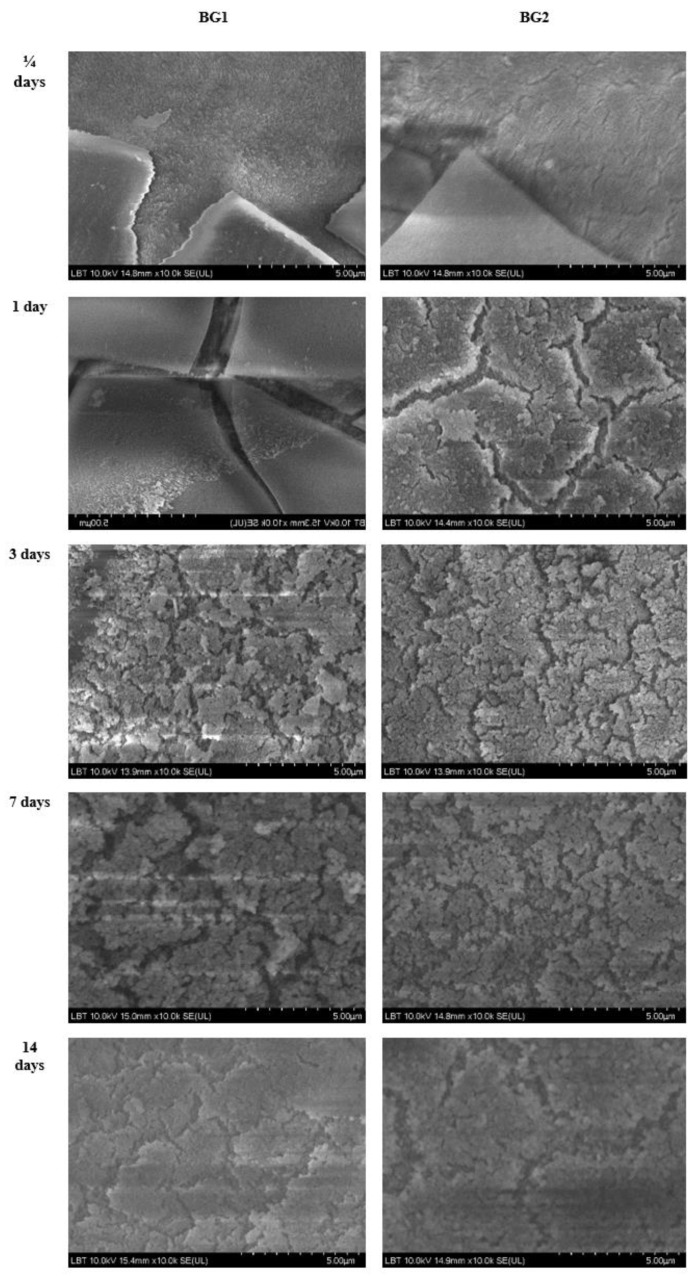
SEM images of samples BG1 (left column) and BG2 (right column) collected at different immersion time intervals (1/4, 1, 3, 7, and 14 days).

**Figure 7 materials-15-09060-f007:**
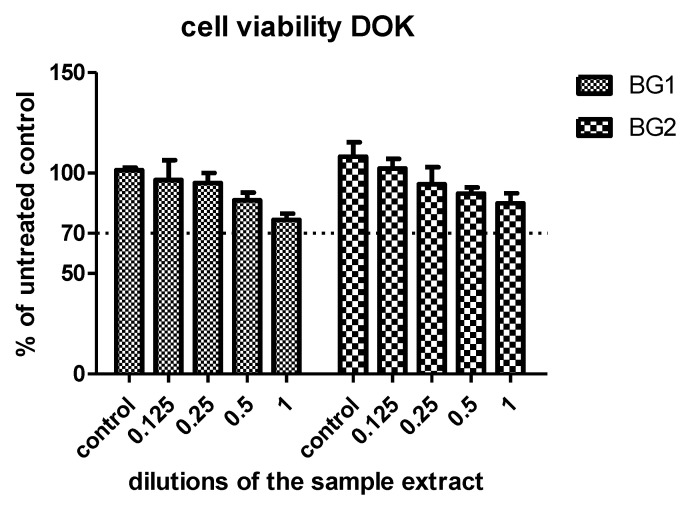
Cell viability of DOK cells exposed to different concentrations of BGs compared with control untreated cells. DOK cells were exposed to dilutions of the sample extract ranging between 0.125–1 and compared with control untreated cells. Data are presented as a mean of OD540 ± SD, *n* = 3 for each sample.

**Figure 8 materials-15-09060-f008:**
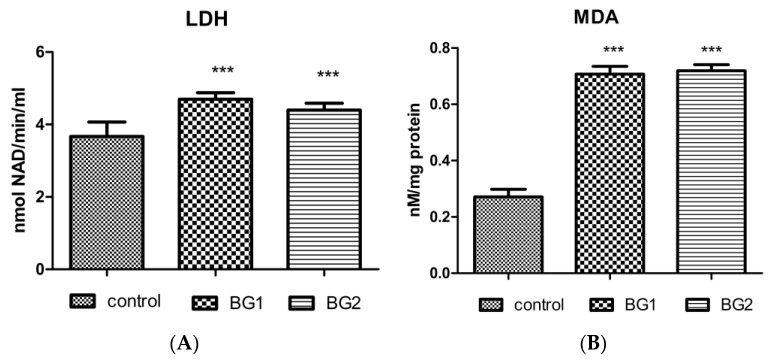
LDH activity and MDA levels in the supernatant of DOK cells exposed to the two BGs compared with untreated control cells: (**A**) LDH activity was significantly increased in the supernatant of DOK after BG exposure, compared with the control (*p* < 0.001); and (**B**) MDA levels after 24 h of exposure to 2 µg/mL BG1 and BG2 were enhanced, compared with control cells (*p* < 0.001). The statistical significance of the difference between exposed and control groups was evaluated by two-way ANOVA and paired Student’s *t*-test followed by the Bonferroni post-hoc test. Data are expressed as the mean of triplicate measurements ± SD. *** *p* < 0.001 vs. control.

**Figure 9 materials-15-09060-f009:**
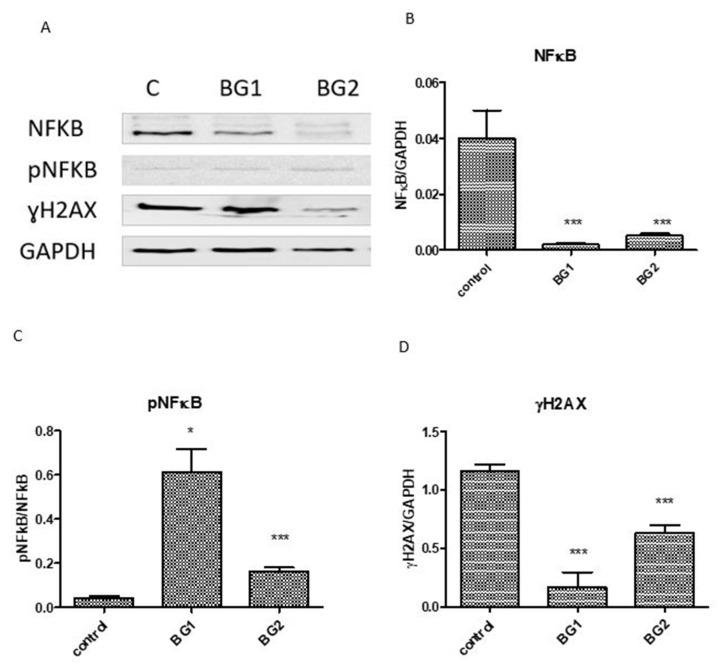
Expression of NF-kB, pNF-KB, and γH2AX in DOK cells exposed to BGs compared with untreated cells: (**A**) image analysis of WB bands was performed by densitometry; the results were normalized to GAPDH (**B**–**D**). The statistical significance between treated cells and the control group was assessed with a two-way ANOVA and paired Student’s *t*-test, followed by the Bonferroni post-hoc test. Each bar represents mean ± standard deviation (*n* = 3); * *p* < 0.05, *** *p* < 0.001 vs. control.

**Table 1 materials-15-09060-t001:** The XPS line positions, FWHM, normalized areas, and At. % composition at ~5.3 nm depth for samples BG1 and BG2. The value marked by * was at the limit of detection and was thus ruled out.

Element	Position (eV)		FWHM (eV)		Area/(RSF*T*MFP) (eV s^−1^)		At. %	
	BG_1_	BG_2_	BG_1_	BG_2_	BG_1_	BG_2_	BG_1_	BG_2_
**B 1*s***	194.25	194.19	2.63	2.66	6391.50	6688.22	31.41	31.13
**Ca 2*p* (3/2)**	349.45	349.47	2.45	2.46	2737.79	2831.42	13.45	13.17
**F 1*s***	687.04	687.05	3.17	2.97	938.84	667.07	4.61	3.10
**O 1*s***	533.65	533.65	3.01	3.03	10260.40	11268.50	50.43	52.44
**V 2*s***	633.04	632.67	2.08	3.17	~3.5 *	29.57	0.07	0.13

**Table 2 materials-15-09060-t002:** Average elemental composition of the two BGs.

Immersion Time		O	B	Ca	F
BG1		43.8	42.2	6.2	3.7
BG2	1/4 d	46.4	52.2	6	4.4
BG1		49.47	50.2	6.2	2.6
BG2	1 d	48.3	50.4	2.9	3.4
BG1		47.5	48.9	4	4.3
BG2	3 d	42.1	46.5	5	3.4
BG1		47.77	45.9	7.23	4.6
BG2	7 d	47.7	44.2	6.8	3.9
BG1		50.4	60.9	3.7	4.3
BG2	14 d	48.1	32.6	7.9	4.5

## Data Availability

Not applicable.

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
