# Peer review of "Characterization and In Vitro Biocompatibility of Two New Bioglasses for Application in Dental Medicine—A Preliminary Study"

_materials, 2022, doi:10.3390/ma15249060_

Round 1

Reviewer 1 Report

The manuscript (materials-2027281) is entitled “Characterization and in vitro biocompatibility of two new bio-glasses for application in dental medicine – a preliminary study”. The authors developed Barium and vanadium containing bio-glasses; further, the synthesized bio-glasses were evaluated for various physicochemical and biological properties. However, to improve the quality of the manuscript, authors must need to address the following comments and queries:

1. The rationale behind the study and the novelty of the work must be described in detail.

2. The Introduction suffers from redundancy. Several points have been repeated again and again.

3. Scanning electron microscopy images and XRD results should be provided for BGs.

4. Image quality in figure 6 is not apt; authors should provide better-quality images.

5. Authors have not mentioned error bars/statistical analysis in some results.

6. In the text mentioned time points are ¼, 1, 3, 7, 14. However, in figure 2 and figure 3, the authors have mentioned more time points. Please explain. Additionally, more time points can be added between day 1 and day 3 to understand the sharp rise in pH values.

7. The in vitro study was performed on DOK cells; It would be better if authors can perform the study on primary cells.

8. Statistical analysis in figure 9C seems incorrect; please check carefully.

9. The discussion is a little poor, authors must correlate the results and discussion for its application in dental medicines. Presently, discussion emphasizes more on bone tissue-related applications.

10. Conclusion section including key insights and future perspectives should be included in the manuscript.

Author Response

Dear Sir,

Attached you will find the answer to the Reviewer 1.

Best regards,

Adriana Filip

Reviewer 2 Report

Abstract:  

- EDAX? it's not EDX? or EDS? and the authors have to explain what the mean of this abbreviation  

- Please add the statistical analysis  

Introduction:  

- please reorganize the introduction in several paragraphs  

- Also correct EDAX  

- Please provide a null hypothesis  

Methods:  

- 110-130: some sentences in red color? why?  

- You have to numer each section: 2.1. , 2.2. .....  

- Line 134: please precise the different time intervals  

- More information about the pH measurement, temperature? Calibrated meter? ....  

- 136-144: which balance? any reference for the sample dimension? or ISO?  

- Line 166: 30Kv??? are you sure?  

- How many samples for each test? please clarify  

Results:  

- Figures 2, 3, 4 and 5: please add the SD to the graph  

- Table 3: please add the SD  

- SEM images: 1/4 BG2 and 7 and 14 days could not be published with this resolution !  

Discussion:  

- Please accept or reject the null hypothesis  

- Please clarify the limitation of the present study  

- Please add the conclusion part  

- Errors in the reference list   I give the author a second chance by major revision to provide their paper

Author Response

Dear Sir,

Attached you will find the answer tot her Reviewer 2.

Best regards,

Adriana Filip

Round 2

Reviewer 1 Report

The authors addressed the comments and queries raised by the reviewer and the manuscript is significantly improved. However, I recommend authors thoroughly check the English language and styles.

Author Response

Response: The manuscript has been corrected from the point of view of the English language (https://www.mdpi.com/authors/english).

Reviewer 2 Report

The revision is good but the problem still that the quality of SEM images is very low

Author Response

Response: I tried to improve the quality of Figure 6 – SEM. Additionally, I ask the Section Managing Editor to help us and to recommend a qualified person from her staff to improve the resolution of SEM figure.